# Stress Responses and Ammonia Nitrogen Removal Efficiency of *Oocystis lacustris* in Saline Ammonium-Contaminated Wastewater Treatment

**DOI:** 10.3390/toxics12050353

**Published:** 2024-05-10

**Authors:** Yuqi Zhu, Yili Zhang, Hui Chen, Lisha Zhang, Chensi Shen

**Affiliations:** 1College of Environmental Science and Engineering, Donghua University, Shanghai 201620, China; 18017207003@163.com (Y.Z.); 17792797372@163.com (Y.Z.); lszhang@dhu.edu.cn (L.Z.); 2Key Laboratory of Agricultural Germplasm Resources Mining and Environmental Regulation of Ningbo City, College of Science and Technology, Ningbo University, Cixi 315302, China; chenhui07@126.com

**Keywords:** saline NH_4_^+^-N wastewater, microalgae treatment technology, *Oocystis lacustris*, NH_4_^+^-N utilization, stress responses

## Abstract

The increasing concern over climate change has spurred significant interest in exploring the potential of microalgae for wastewater treatment. Among the various types of industrial wastewaters, high-salinity NH_4_^+^-N wastewater stands out as a common challenge. Investigating microalgae’s resilience to NH_4_^+^-N under high-salinity conditions and their efficacy in NH_4_^+^-N utilization is crucial for advancing industrial wastewater microalgae treatment technologies. This study evaluated the effectiveness of employing nitrogen-efficient microalgae, specifically *Oocystis lacustris*, for NH_4_^+^-N removal from saline wastewater. The results revealed *Oocystis lacustris*’s tolerance to a Na_2_SO_4_ concentration of 5 g/L. When the Na_2_SO_4_ concentration reached 10 g/L, the growth inhibition experienced by *Oocystis lacustris* began to decrease on the 6th day of cultivation, with significant alleviation observed by the 7th day. Additionally, the toxic mechanism of saline NH_4_^+^-N wastewater on *Oocystis lacustris* was analyzed through various parameters, including chlorophyll-a, soluble protein, oxidative stress indicators, key nitrogen metabolism enzymes, and microscopic observations of algal cells. The results demonstrated that when the *Oocystis lacustris* was in the stationary growth phase with an initial density of 2 × 10^7^ cells/L, NH_4_^+^-N concentrations of 1, 5, and 10 mg/L achieved almost 100% removal of the microalgae on the 1st, 2nd, and 4th days of treatment, respectively. On the other hand, saline NH_4_^+^-N wastewater minimally impacted photosynthesis, protein synthesis, and antioxidant systems within algal cells. Additionally, NH_4_^+^-N within the cells was assimilated into glutamic acid through glutamate dehydrogenase-mediated pathways besides the conventional pathway involving NH_4_^+^-N conversion into glutamine and assimilation amino acids.

## 1. Introduction

Water eutrophication, resulting from an excess of nitrogen, phosphorus, and other nutrients in water, poses a threat to water resources worldwide [1,2]. Ammonia nitrogen (NH_4_^+^-N) is a typical pollutant that contributes to eutrophication in natural water bodies and presents a risk to aquatic ecosystems [3]. In 2020, global ammonia production increased by 2.9 million tons, reaching a total supply of 185 million tons [4]. Besides wastewater from residential and agricultural sources, emissions of NH_4_^+^-N from industrial activities must not be disregarded [5]. Typically, industries such as food processing, rubber processing, textile dyeing and printing, leather manufacturing, fertilizer production, and others, discharge high levels of NH_4_^+^-N concentration into water [6]. As an illustration, in China, the “Second National Pollution Source Census Bulletin” released in June 2020 revealed that industrial wastewater in China discharged 44,500 tons of NH_4_^+^-N in 2017 [7]. In general, governments worldwide have set strict environmental standards for NH_4_^+^-N [8]. For example, the United States Environmental Protection Agency (U.S. EPA) recommends that in freshwater systems, acute one-hour exposures should not exceed 17 mg/L (pH = 7.0 and T = 20 °C) and also regulates the chronic exposures [9]. Facing the urgent challenges of global warming, it is crucial to seek methods for treating NH_4_^+^-N in industrial wastewater that are efficient, cost-effective, and characterized by low energy consumption [10,11].

Currently, a variety of NH_4_^+^-N removal technologies are available, including physical methods like air stripping, ion exchange, and adsorption; chemical methods such as chemical precipitation and breakpoint chlorination; and biological methods like nitrification, denitrification, partial nitrification, and anaerobic ammonium oxidation [12,13,14,15]. In comparison, the growing concern about climate change and water conservation has drawn significant attention from researchers to investigate the potential of microalgae for biological carbon fixation and wastewater treatment [16,17,18]. However, microalgae technology for industrial wastewater treatment is not yet fully developed, and there are still areas of concern that require further research. On one hand, while microalgae primarily utilize NH_4_^+^-N as their nitrogen source, excessive concentrations have been observed to induce toxic effects, thereby inhibiting growth [19]. On the other hand, microalgae are susceptible to the effects of coexisting pollutants in industrial wastewater, which can inhibit their growth. For example, the presence of salt compounds in industrial wastewater can diminish algal activity due to high osmotic pressure, thus impacting the treatment efficacy for NH_4_^+^-N [20,21]. While certain types of microalgae can tolerate fluctuations in salinity, exposure to salinity stress often leads to reduced biomass productivity, mainly due to the significant energy requirements for osmoregulation [22].

*Oocystis lacustris*, a member of the *Oocystis genus*, is commonly distributed across various water bodies, particularly thriving in freshwater ecosystems, where it dominates the planktonic community in small lakes and ponds [23]. Additionally, previous research has found that *Oocystis lacustris* can thrive in food waste digestate with high NH_4_^+^-N concentrations, and researchers have identified it as one of the prevalent algae species in surface water with elevated salinity levels [24]. As such, *Oocystis lacustris* holds promising potential for the treatment of saline NH_4_^+^-N wastewater. Nonetheless, it is essential to conduct laboratory experiments to evaluate the toxicity of NH_4_^+^-N and to explore the tolerance of *Oocystis lacustris* to saline industrial wastewater, as well as assess its effectiveness in removing NH_4_^+^-N.

Therefore, with the goal of further advancing the application of microalgae technologies in industrial wastewater treatment, this study simulated the characteristics of saline NH_4_^+^-N wastewater and investigated the growth of *Oocystis lacustris* in the wastewater, as well as its efficiency in removing NH_4_^+^-N. Importantly, this research analyzed the physiological status, lipid accumulation, key enzyme activity, and changes in the antioxidant system of *Oocystis lacustris* during the treatment of saline NH_4_^+^-N wastewater, contributing to a more comprehensive understanding of microalgae-based saline NH_4_^+^-N wastewater treatment technology. Furthermore, the potential mechanisms involved in the removal of NH_4_^+^-N have been examined, aiming to provide valuable guidelines for the treatment of saline NH_4_^+^-N wastewater using microalgae.

## 2. Materials and Methods

### 2.1. Instruments and Reagents

The experimental devices used in this study included an intelligent light incubator (GXZ, Ningbo Jiangnan Instrument Factory, Ningbo, China), algal cell counter (IA1000, Shanghai Ruiyu Biotechnology Co., Ltd., Shanghai, China), spectral color illuminometer (OHSP-350P, Hangzhou Hopoo Light & Color Technology Co., Ltd., Hangzhou, China), UV–visible spectrophotometer (UV-1900, Shimadzu, Kyoto, Japan), vertical pressure steam sterilizer (LDZM-40KCS, Shanghai Shenan Medical Equipment Factory, Shanghai, China), UV sterilization bench (VD-850, Shanghai Dingke Scientific Instrument Co., Ltd., Shanghai, China), Heraeus Multifuge X1 refrigerated centrifuge (Thermo Fisher Scientific, Waltham, MA, USA), and ultrasonic cell disruptor (JY92-II, Ningbo Xinzhi Biotechnology Co., Ltd., Ningbo, China).

The reagents used in this study, including sodium sulfate (Na_2_SO_4_) and phosphate-buffered saline (PBS) purchased from Shanghai Aladdin Biochemical Technology Co., Ltd. (Shanghai, China), and ammonium sulfate [(NH_4_)_2_SO_4_] obtained from China National Pharmaceutical Group Corporation (Beijing, China), were all of analytical grade. Additionally, a BG11 medium was used which contained the following components: NaNO_3_ (1.5 g/L); K_2_HPO_4_·3H_2_O (40 mg/L); MgSO_4_·7H_2_O (75 mg/L); CaCl_2_·2H_2_O (36 mg/L); Na_2_CO_3_ (20 mg/L); FeCl_3_·6H_2_O (3.15 mg/L); citric acid (6 mg/L); and 1 mL of microelements composed of H_3_BO_3_ (2.86 mg/L), MnCl_2_·4H_2_O (1.81 mg/L), ZnSO_4_·7H_2_O (0.22 mg/L), Na_2_MoO_4_·2H_2_O (0.39 mg/L), CuSO_4_·5H_2_O (0.08 mg/L), and Co(NO_3_)_2_·6H_2_O (0.05 mg/L) in 1000 mL, and the pH was adjusted to 7.1 using 0.1 mol/L HCl and 0.1 mol/L NaOH.

### 2.2. Algal Strain and Culture Conditions

The algal strain, *Oocystis lacustris* (FACHB-2069), used in this study was obtained from the Freshwater Algal Culture Collection at the Institute of Hydrobiology, Chinese Academy of Sciences (Wuhan, China). The propagation of the strain involved the preparation and sterilization of the BG11 medium through autoclaving at 100 kPa and 121 °C for 20 min. The algal inoculum was introduced into the culture medium at a volume ratio of 1:5, and the resulting mixture was then placed in an intelligent light incubator. They were maintained at 25 ± 1 °C and illuminated by 60 μmol photons m^−2^ s^−1^ from daylight-type fluorescent lamps using a light:dark photoperiod of 12:12. Irradiance was measured by a spectral color illuminometer. The conical flasks were manually agitated three times daily at regular intervals to prevent algal cell sedimentation.

### 2.3. The Effect of High Concentrations of Na_2_SO_4_ on Algal Growth

Na_2_SO_4_ is a common dyeing accelerant for reactive dyes, favored for its superior performance over NaCl [25]. The concentration of SO_4_^2−^ in dyeing wastewater typically ranges from 200 to 5000 mg/L, sometimes exceeding 10,000 mg/L [25,26]. Generally, wastewater is considered “high-salinity” when the concentration of inorganic salts ranges from 1 to 3.5% *w*/*w*. Therefore, this study investigated the effects of Na_2_SO_4_ concentrations ranging from 5 to 20 g/L (equivalent to approximately 3380 to 13,520 mg/L of SO_4_^2−^) on microalgal growth. Accordingly, the BG11 medium (the conductivity was 0.19 S/m) was prepared using Na_2_SO_4_ as an inorganic salt source at concentrations of 5, 10, 15, and 20 g/L (the conductivities were 0.78 S/m, 1.22 S/m, 1.64 S/m, and 2.07 S/m), respectively. The algal cultures in the exponential growth phase (on the 4th to 6th day after inoculation) were utilized. Considering the requirements for the hydraulic retention time (HRT) and treatment efficiency in the wastewater treatment, the initial concentration of microalgae was established at 8 × 10^5^ cells/L, surpassing the standards outlined by the OECD (2011) Test No. 201 and other pertinent guidelines for algal growth inhibition tests [27]. A control group using the standard BG11 medium was established for comparative analysis. During the tests, a 50 mL microalgal suspension was cultured in a 100 mL conical flask sealed with a breathable sealing film. The cultivation conditions for *Oocystis lacustris*, including illumination, temperature, and agitation, were consistent with the methods described in Section 2.2 regarding the cultivation of algal strains. This experimental section was conducted three times to ensure the reproducibility of the results. All tests included triplicates of each Na_2_SO_4_ concentration and five replicates of the control sample. The samples were taken every 24 h to measure the algal cell density (Section 2.5.1).

### 2.4. NH_4_^+^-N Removal by Oocystis lacustris in Simulated Saline Wastewater

In the textile printing and dyeing process, printing necessitates the utilization of significant quantities of urea, which undergoes transformation into NH_4_^+^-N during the biological treatment of wastewater. Additionally, the post-finishing treatment of cotton fabrics frequently entails the application of liquid ammonia to enhance their sheen and durability, thereby directly contributing to the generation of NH_4_^+^-N [26]. The concentration of NH_4_^+^-N in textile printing and dyeing wastewater ranges from 0.5 to 75 mg/L [26]. Therefore, nitrogen components were removed from the BG11 medium, and the Na_2_SO_4_ concentration was set to 10 g/L. NH_4_^+^-N concentrations of 1, 5, 10, 30, and 50 mg/L were then prepared to simulate saline NH_4_^+^-N wastewater in our study. The BG11 medium containing 10 g/L Na_2_SO_4_ served as the control group. *Oocystis lacustris* cultures in the exponential growth phase were inoculated into the simulated wastewater at an initial cell density of 8 × 10^5^ cells/mL, referred to as the EX phase *Oocystis lacustris* treatment group. To investigate the effect of *Oocystis lacustris* during stationary growth phases (on the 8th to 10th day after inoculation) on NH_4_^+^-N removal, the microalgae in the stationary growth phase were inoculated into the simulated wastewater at an initial cell density of 2 × 10^7^ cells/mL. This group was referred to as the STA phase *Oocystis lacustris* treatment group. The cultivation conditions, including illumination, temperature, and agitation, remained consistent with the methods outlined in Section 2.2. This experimental section was also repeated three times to ensure result reproducibility. Each test included triplicates of each NH_4_^+^-N concentration and five replicates of the control sample. Given that enzyme assays require a larger quantity of algal cells, an additional three replicates were added to the treatment groups that needed enzyme activity measurement. The samples were collected at fixed intervals to measure algal cell density (Section 2.5.1), NH_4_^+^-N concentration (Section 2.5.2), and physiological parameters (Section 2.5.3). Additionally, under the conditions of pH = 7.1 and the temperature = 25 °C, the proportion of NH_4_^+^ unionized as NH_3_ was less than 1%.

### 2.5. Analytical Methods

#### 2.5.1. Algal Cell Density

First, 1 mL of a microalgal solution was mixed with 10 μL of Lugol’s solution for fixation. Subsequently, 30 µL of the mixture was transferred to a cell counting chamber, and the algal cell density was measured using the Countstar algal cell counter. During the counting process, each sample was measured three times for replicability. The percent inhibition in yield (%*I_y_*) was calculated for each treatment replicate as follows [27]:(1)$$\%I_{y}=\frac{\left(Y_{c}-Y_{T}\right)}{Y_{c}}\times 100$$
where %*I_y_* represents the percent inhibition of yield, *Y_c_* represents the mean value for yield in the control group, and *Y_T_* represents the value for yield for the treatment replicate. The yield is calculated as the biomass at the end of the test minus the starting biomass for each single vessel of controls and treatments.

#### 2.5.2. NH_4_^+^-N Concentrations

NH_4_^+^-N concentrations were determined via Nessler’s reagent method, with each sample being measured three times to ensure reproducibility [28].

#### 2.5.3. Physiological Parameters of the Microalgae

For the determination of chlorophyll-a (*Chl-a*), the alga sample (10 mL) was collected by centrifugation with 10,000 rpm for 10 min. The collected algal cells were frozen and thawed four times in the dark. Next, the solids were steeped in an acetone 80% solution at 4 °C for 24 h. At last, the acetone solution containing chlorophyll-a was centrifuged again with 10,000 rpm for 10 min and the supernatant was measured by a UV–visible spectrophotometer with OD_630_, OD_647_, OD_664_, and OD_750_ [29,30].
(2)$$Chl-a=\frac{11.5(OD_{664}-OD_{750})-1.54(OD_{647}-OD_{750})-0.08(OD_{630}-OD_{750})V_{1}}{V}$$
where *Chl-a* is the chlorophyll-a concentration of the sample (mg/L), *V*_1_ is the volume of the sample after extraction (mL), and *V* is the volume of the sample (mL).

To investigate soluble protein, superoxide dismutase (SOD), catalase (CAT), malondialdehyde (MDA), triglycerides (TAG), nitrate reductase (NR), and glutamine synthetase (GS) within microalgae, algal cells were disrupted first. Algal cells were collected from the microalga suspension (40 mL) by centrifugation (10,000 rpm) at 4 °C for 10 min. The cell pellets were washed by pre-cooled phosphate-buffered saline (PBS) solution (50 mmol/L, pH 7.0) twice and re-suspended in 6 mL of a PBS solution. Cell homogenization was conducted by ultrasonic cell disintegration at 650 W for 25 cycles (ultrasonic time: 3 s; rest time: 9 s) in an ice bath. The disrupted cells were then centrifuged at 10,000 rpm for 10 min at 4 °C, and the supernatant was collected and kept on ice for soluble protein content, MDA content, TAG content, and enzymes activity measurements. 

The assay kits for measuring soluble protein (product no. QYS-237014), SOD (product no. QYS-23028), CAT (product no. QYS-23030), MDA (product no. QYS-23036), TAG (product no. QYS-234006), NR (product no. QYS-232003), and GS (product no. QYS-232017), were purchased from Qiyi Biological technology (Shanghai) Co., Ltd. (Shanghai, China). Soluble protein content was determined using the bicinchoninic acid (BCA) method at 562 nm wavelength using a UV–visible spectrophotometer. SOD activity was evaluated utilizing the nitro-blue tetrazolium dye (NBT) method at a 560 nm wavelength using a UV–visible spectrophotometer. CAT activity was assessed based on H_2_O_2_ decomposition, measured at 240 nm with a UV–visible spectrophotometer. The MDA level was determined by the formation of the MDA-TBA adduct formed by the reaction of MDA and thiobarbituric acid (TBA) under high temperature (90–100 °C), with absorbance measured at 535 nm using a UV–visible spectrophotometer. These enzyme activities and the MDA content were quantified by protein concentration. The enzyme activity unit (U) was defined as the amount of enzyme required to catalyze 1 μmol of substrate per minute. The TAG content was determined using a UV–visible spectrophotometer at 420 nm. The colorimetric principle was as follows: TAG was saponified by KOH to hydrolyze into glycerol and fatty acids. Glycerol was oxidized to formaldehyde by periodic acid. In the presence of chloride ions, formaldehyde condensed with acetone to produce a yellow substance. The NR activity was determined by monitoring the oxidation of NADH at 340 nm (UV-visible spectrum), dependent on the reduction of NO_3_^−^ to NO_2_^−^. The GS activity was determined by the reverse-glutamyl transferase reaction, which measured the formation of glutamyl hydroxamate using UV–visible spectrophotometer at 540 nm. One unit of enzymatic activity (U) was defined as the formation of 1 μmol of *γ*-glutamyl hydroxamate per min. These enzyme activities and TAG content were quantified by the algal cell concentration. The assays mentioned above were conducted following the user manual instructions of the assay kits, with each sample subjected to triplicate measurements.

#### 2.5.4. Subcellular Structures of the Microalgae

The *Oocystis lacustris* cells were initially fixed in 3% glutaraldehyde in a 0.1 mol/L cacodylate buffer, followed by fixation in 1% aqueous osmium tetroxide in a 0.1 mol/L cacodylate buffer. After dehydration in acetone and embedding in Spurr’s resin, ultrathin sections were prepared and stained with uranyl acetate and lead citrate [31]. The sections were then observed under a Hitachi HC-1 transmission electron microscope (TEM) at an accelerating voltage of 80 kV.

#### 2.5.5. Data Processing and Analysis

Data analysis was conducted using the SPSS software statistical package (SPSS version 17.0 for Windows). The mean and standard error of the mean (S.E.M) were calculated for each parameter. The results were compared to determine the toxic effects by a one-way analysis of variance (ANOVA) and graphically presented using GraphPad Prism version 6 (GraphPad Software Inc., La Jolla, CA, USA). The significance level was set at *p*-value 0.05.

## 3. Results and Discussion

### 3.1. Influence of Different Na_2_SO_4_ Concentrations on the Growth of Oocystis lacustris

Figure 1 shows the growth of *Oocystis lacustris* at different concentrations of Na_2_SO_4_. The cell concentration of *Oocystis lacustris* increased across all tested Na_2_SO_4_ concentrations, despite significant differences in the growth rates (Figure 1a). On the 7th day of cultivation, the cell concentration of *Oocystis lacustris* in the 20 g/L Na_2_SO_4_ group was only half that of the control group. The growth inhibition (Figure 1b) shows that the highest inhibition occurred on the 1st day of cultivation under saline conditions, likely due to the significant impact of Na_2_SO_4_ exposure on algal growth. On the 2nd day, there was partial relief in growth inhibition. However, except for the 5 g/L concentration, Na_2_SO_4_ continued to inhibit *Oocystis lacustris* growth, reaching a second peak of inhibition by the 5th day of cultivation. In addition to altering osmotic pressure, Na_2_SO_4_ may also compete with structurally similar nutrients for transport proteins, thereby reducing the ability of algal cells to acquire essential other nutrients for growth, such as selenite and molybdate ions [32,33,34]. It was also observed that as the algal cell concentration increased, the standard deviation among replicate samples widened. This could be attributed to counting errors caused by the aggregation of *Oocystis lacustris*. To ensure the reliability of observation, two additional replicate experiments were conducted, as depicted in Appendix A. Although variations in numerical values existed in algal cell concentration between different experimental batches, the overall inhibitory trend remained consistent. Due to the results indicating that under treatment with 10 g/L Na_2_SO_4_, although *Oocystis lacustris* exhibited significant growth inhibition, recovery was observed by the 7th day. Furthermore, the concentration of 10 g/L Na_2_SO_4_ closely resembles the characteristics of actual high-salinity wastewater. Subsequent experiments in our study were conducted using 10 g/L Na_2_SO_4_ to simulate saline wastewater.

### 3.2. Effects of Oocystis lacustris on NH_4_^+^-N Removal

Figure 2a shows the NH_4_^+^-N removal efficiency of *Oocystis lacustris* with an initial cell density of 8 × 10^5^ cells/mL (EX phase) at different concentrations of NH_4_^+^-N. *Oocystis lacustris* exhibited high removal efficiencies at low concentrations of NH_4_^+^-N, achieving almost 100% removal rate by the 2nd day for 1 mg/L NH_4_^+^-N and by the 7th day for 5 mg/L NH_4_^+^-N. However, as the NH_4_^+^-N concentration exceeded 10 mg/L, the NH_4_^+^-N removal efficiency by the algae decreased, reaching 70% on the 11th day. As the NH_4_^+^-N concentration further increased to 30 mg/L, the removal efficiency of the algae remained low. This phenomenon was closely related to the growth state of algal cells (Figure 2b,c), which illustrated the growth of algal cells and the corresponding growth inhibition over time. During the 11-day growth period, the cell density of *Oocystis lacustris* in the control group increased from 8 × 10^5^ to 4.6 × 10^7^ cells/mL. The presence of NH_4_^+^-N obviously inhibited the growth of *Oocystis lacustris*. On the 11th day of the NH_4_^+^-N treatment experiment, in the treatment groups with NH_4_^+^-N concentrations ranging from 1 to 50 mg/L, the inhibition rate of microalgae ranged from 23.6% to 77.7%. When the NH_4_^+^-N concentrations were 1 mg/L and 5 mg/L, although the removal rate of NH_4_^+^-N could reach nearly 100%, the growth inhibition rate of *Oocystis lacustris* still ranged from 23.6% to 45.4%. The consistent phenomena observed from replicate experiments are illustrated in Appendix A. This suggests that *Oocystis lacustris* in the EX phase was sensitive to the NH_4_^+^-N concentration in saline wastewater. Previous studies have indicated that high NH_4_^+^-N levels mainly disrupt the normal transmembrane proton gradient from adenosine triphosphate (ATP) to ADP in algal chloroplasts, inhibiting photosynthesis. This disruption may occur through mechanisms such as damage to protein subunits in chloroplasts, thereby hindering the function of photosystem II (PSII) and inhibiting cell growth [35,36,37].

Figure 3a illustrates the NH_4_^+^-N removal efficiency of *Oocystis lacustris* with an initial cell density of 2 × 10^7^ cells/mL (STA phase) at different concentrations of NH_4_^+^-N. Compared with the EX phase, the NH_4_^+^-N removal efficiency of *Oocystis lacustris* in the STA phase was improved. NH_4_^+^-N concentrations of 1, 5, and 10 mg/L resulted in almost complete removal by *Oocystis lacustris* on the 1st, 2nd, and 4th days of treatment, respectively. As the NH_4_^+^-N concentration increased to 30 and 50 mg/L, its removal efficiency remained low, with NH_4_^+^-N removal rates of 35.2% and 16.2% on the 8th day of treatment, respectively. Additionally, compared to the growth phase of EX, *Oocystis lacustris* in the STA phase exhibited a slower rate of cell proliferation during NH_4_^+^-N removal. By the 8th day of cultivation, the cell density of *Oocystis lacustris* in the control group had only increased 2.5-fold. The slower growth rate resulted in the peak inhibition of NH_4_^+^-N occurring on the first day of cultivation. Subsequently, the growth inhibition remained stable, with the growth inhibition of algal cells in saline NH_4_^+^-N environments all below 50%. The consistent observations from replicate experiments are depicted in Appendix A. It indicates that microalgae in the STA phase with high cell densities might be more advantageous for treating saline NH_4_^+^-N wastewater. Despite initially experiencing some toxicity impact during treatment, the microalgae demonstrated rapid stabilization in growth. Moreover, they exhibited high efficiency in removing NH_4_^+^-N.

Table 1 displays the NH_4_^+^-N removal efficiency achieved by typical microalgae. Given the diversity in species, culture conditions, and wastewater compositions, the NH_4_^+^-N removal efficiency varies widely, ranging from 31% to over 97%. Such variability poses significant challenges for microalga-based wastewater treatment, particularly when handling large wastewater volumes. Moreover, there is a scarcity of studies examining NH_4_^+^-N removal efficiency under high-salinity conditions. Therefore, this study aimed to investigate the NH_4_^+^-N removal efficiency of *Oocystis lacustris* under high concentrations of Na_2_SO_4_. While the removal efficiency may appear lower compared to the reported algal strains such as *Chlorella* sp., *Chlorococcum* sp., *Parachlorella kessleri*, and others, we explored the stress induced by high concentrations of Na_2_SO_4_. To delve deeper into the physiological responses of *Oocystis lacustris* to saline NH_4_^+^-N wastewater treatment, subsequent sections will undertake analyses of chlorophyll-a content, oxidative stress indicators, key nitrogen metabolism enzymes, and microscopic observations of algal cells. These investigations aim to lay a foundation for the potential application of *Oocystis lacustris* in treating saline NH_4_^+^-N wastewater.

### 3.3. Variations in Chlorophyll-a and Soluble Protein Contents of Oocystis lacustris

Figure 4 illustrates the changes in chlorophyll-a and soluble protein contents in *Oocystis lacustris* cells treated with NH_4_^+^-N on the 8th day with the presence of 10 g/L Na_2_SO_4_. In the EX phase group of microalgae, when the NH_4_^+^-N concentration exceeded 10 mg/L, there was a significant decrease in chlorophyll-a content, reaching only one-fifth of the level observed in the control group as the NH_4_^+^-N concentration reached 30 mg/L (Figure 4a). The treatment group with NH_4_^+^-N concentration at 5 mg/L was an outlier. As evident from the results of the two additional repeated experiments (Appendix A), at this concentration, the effect of NH_4_^+^-N on chlorophyll-a content was not significant. Changes in the soluble protein content of *Oocystis lacustris* cells were consistent with those of chlorophyll-a (Figure 4b). On the 8th day of treatment with 30 and 50 mg/L NH_4_^+^-N, the soluble protein content in the algal cells significantly decreased, reaching only 36% of the level observed in the control group. These reductions may be attributed to the detrimental effects of high NH_4_^+^-N concentrations on the oxygen-evolving complex (OEC) of algal cells, leading to reduced OEC activity. Additionally, NH_4_^+^-N diffusion within cells altered the pH of the thylakoid lumen, resulting in decreased ATP levels, thereby affecting chlorophyll-a synthesis [44,45].

In comparison, *Oocystis lacustris* in the STA phase exhibited stronger resistance to the toxic impacts of saline NH_4_^+^-N wastewater. As the NH_4_^+^-N concentration exceeded 30 mg/L, there was only a 29.3% decrease in chlorophyll-a content. On the other hand, with an increasing NH_4_^+^-N concentration, the soluble protein content in the algal cells peaked at 0.108 mg/mL in the 10 mg/L NH_4_^+^-N treatment group (Figure 4d). Subsequently, the soluble protein content gradually decreased, reaching a minimum of 0.046 mg/mL in the 50 mg/L NH_4_^+^-N group. Research reports have indicated that the accumulation of soluble proteins in microalgae can enhance cell water retention capacity and protect vital compounds within the cell membrane [46,47]. Therefore, at lower concentrations of NH_4_^+^-N, the synthesis of soluble proteins in algal cells increased, whereas exposure to high concentrations of NH_4_^+^-N inhibited and significantly damaged the synthesis of soluble proteins in algal cells.

### 3.4. Oxidative Stress Status of Oocystis lacustris

The oxidative stress induced by NH_4_^+^-N and Na_2_SO_4_ may affect the growth status of *Oocystis lacustris* and the efficiency of NH_4_^+^-N removal. MDA, a product of lipid peroxidation, is generally considered a primary biomarker for assessing oxidative stress intensity [48,49,50]. Figure 5a,d shows the accumulation of MDA in algal cells during the removal of NH_4_^+^-N. For the microalgae in the EX phase (Figure 5a), the MDA content in *Oocystis lacustris* cells remained relatively stable in the 10 mg/L NH_4_^+^-N treatment group compared with the control group. However, at a high NH_4_^+^-N concentration (50 mg/L), the MDA content rapidly increased, indicating strong oxidative stress and significant oxidative damage to algal cells [51,52]. At the same time, the MDA content of *Oocystis lacustris* in the STA phase remained relatively stable (Figure 5d). Figure 5b,c,e,f shows the response of the antioxidant system in *Oocystis lacustris* through the evaluation of the concentrations of SOD and CAT in algal cells. SOD was considered the primary enzyme in the antioxidant system. Moreover, SOD catalyzed the dismutation reaction of superoxide radicals into hydrogen peroxide and oxygen molecules, which were further decomposed into water and molecular oxygen by enzymes such as CAT [53,54,55]. In the EX phase (Figure 5b,c), microalgae showed increased SOD and CAT activities with rising NH_4_^+^-N concentrations. By the 8th day of treatment with 50 mg/L NH_4_^+^-N, the SOD activity doubled, while the CAT activity increased nearly fourfold compared to the control group. These changes, along with the replicated experimental results (Appendix A), indicate that during the EX phase, *Oocystis lacustris* was highly sensitive to the toxic effects of NH_4_^+^-N, leading to significant oxidative stress. In contrast, SOD and CAT activities within *Oocystis lacustris* cells during the STA phase showed only slight increases, even in the presence of 50 mg/L NH_4_^+^-N, suggesting slight oxidative stress.

### 3.5. Subcellular Structures of Oocystis lacustris

Figure 6 shows the cell morphology of *Oocystis lacustris* in the EX phase. In the control group, algal cells exhibited an elliptical or circular shape, with multiple layers of cell walls and well-organized chloroplasts. In each algal cell, there is a single pyrenoid enclosed by a homogeneous matrix and surrounded by a thick starch sheath. When treated with 10 mg/L NH_4_^+^-N, the number of starch granules in the algal cells decreased (Figure 6b1–b3). This reduction may be attributed to the high metabolic state induced by NH_4_^+^-N absorption with the presence of Na_2_SO_4_, leading to the degradation and depletion of starch granules. When exposed to 50 mg/L NH_4_^+^-N (Figure 6c1–c3), the starch sheath thickness in algal cells visibly decreased, with widened channels. Moreover, chloroplasts displayed overlapping arrangements instead of being organized in parallel, indicating possible chloroplast damage. Figure 7 shows the cell morphology of *Oocystis lacustris* in the STA phase. In both the control group and the 10 mg/L NH_4_^+^-N treatment group, the main organelles of algal cells did not undergo visible changes. But similar to the algal cell in the EX phase, the number of starch grains decreased in the presence of NH_4_^+^-N. Additionally, a key morphological feature observed was the presence of some cells enveloped in a pectinaceous gelatinous sheath during growth. Sometimes, multiple algal cells aggregated into colonies, encased within a gelatinous sheath (Appendix A) [56,57]. When exposed to 50 mg/L NH_4_^+^-N, the gelatinous sheath ruptured, potentially increasing the susceptibility of the algal cells to external disruptions. This phenomenon indirectly indicates why *Oocystis lacustris* in the STA phase exhibits greater resistance to saline NH_4_^+^-N stress.

### 3.6. Removal Mechanism and Limiting Factors of Oocystis lacustris for NH_4_^+^-N

Microalgae mainly utilized NH_4_^+^-N by synthesizing glutamine from two molecules of α-ketoglutarate using GS-glutamate synthase (GS-GOGAT). The synthesized glutamine was then assimilated into amino acids, which were transported to various organelles for protein synthesis, nucleic acid synthesis, and other life activities. In NO_3_^−^-N utilization, microalgae initially reduced NO_3_^−^-N to NO_2_^−^-N using NR. NO_2_^−^-N was then transferred to the chloroplast and reduced to NH_4_^+^-N using nitrite reductase. Compared to NO_3_^−^-N and NO_2_^−^-N, microalgae tend to preferentially utilize NH_4_^+^-N directly, but NH_4_^+^-N serves both as an essential nutrient and a toxic compound for microalgae. Li found that the GS activity increased with low concentrations of NH_4_^+^-N but decreased as the NH_4_^+^-N concentration increased [58]. This result aligns with our research findings (Figure 8a), showing an increase in GS activity when *Oocystis lacustris* in the EX phase was treated with 10 mg/L NH_4_^+^-N, but a significant decrease when treated with 50 mg/L NH_4_^+^-N. However, *Oocystis lacustris* in the STA phase exhibited no significant fluctuations in GS activity upon treatment with NH_4_^+^-N, indicating minimal disruption to its GS-GOGAT pathway. In addition to the GS-GOGAT pathway, microalgae might also utilize NH_4_^+^-N through the glutamate dehydrogenase (GDH) pathway. Tian et al. found that increasing NH_4_^+^-N concentration, GS activity decreased while GDH expression levels increased [59]. In this pathway, α-ketoglutarate was converted to glutamate salt by GDH [60,61]. 

Additionally, this study investigated alterations in NR activity during the NH_4_^+^-N removal process. Both *Oocystis lacustris* in the EX and STA phases exhibited decreased NR activity when treated with NH_4_^+^-N. Compared to the control group, NR activity in algal cells subjected to 50 mg/L NH_4_^+^-N decreased by 48.9% and 19.0%, respectively (Figure 8b,e). These findings align with those of Tian et al., who observed a significant downregulation of nitrate transport proteins (NTR2 and NTR3) when NH_4_^+^-N was utilized as a sole nitrogen source or in combination with NO_3_^−^-N, indicating the inhibition of nitrate transport protein synthesis by NH_4_^+^-N [59].

Further, TAG is the energy molecule stored as oil droplets within cells, crucial for adapting to adverse environmental conditions and maintaining cell survival processes. The TAG content within *Oocystis lacustris* in the EX phase (Figure 8c) markedly decreased when the NH_4_^+^-N concentration reached 50 mg/L. This decline can be attributed to the damage caused by the elevated NH_4_^+^-N concentration to chloroplasts, resulting in disrupted TAG synthesis and diminished TAG content [62,63]. For the *Oocystis lacustris* in the STA phase, the influence on cellular TAG content was comparatively minimal (Figure 8f), suggesting a potentially heightened resistance capability.

In summary, combining the results of the replicated experiments (Appendix A), *Oocystis lacustris* cultured in simulated saline NH_4_^+^-N wastewater exhibited greater stability during the STA phase and demonstrated enhanced resistance to NH_4_^+^-N stress following stabilization. 

Although NH_4_^+^-N utilization was more direct than NO_3_^−^-N, high concentrations of NH_4_^+^-N can exert toxic effects on algal cells. This induced oxidative stress in algal cells, leading to an imbalance in antioxidant systems and lipid damage. Additionally, high NH_4_^+^-N concentrations could disrupt the OEC of the algal cells and affect chlorophyll-a synthesis, thereby influencing the photosynthetic system of the algal cell. Moreover, the exposure of algal cells to high NH_4_^+^-N concentrations in saline wastewater could lead to lower activities of nitrogen metabolism enzymes such as GS and NR. When using *Oocystis lacustris* to treat saline wastewater containing NH_4_^+^-N, it is important to monitor the stability of microalgae growth and avoid excessively high concentrations of NH_4_^+^-N.

## 4. Conclusions

This study highlights the impact of high concentrations of Na_2_SO_4_ on the growth of *Oocystis lacustris*, demonstrating its tolerance up to 5 g/L, with an observed alleviation of growth inhibition at concentrations of 10 g/L within a week. In the STA phase, *Oocystis lacustris* (2 × 10^7^ cells/mL) exhibited relatively higher efficiency in NH_4_^+^-N removal in the presence of 10 g/L Na_2_SO_4_, achieving almost 100% removal rate for 10 mg/L NH_4_^+^-N by the 4th day of treatment. Additionally, it displayed robust resistance to the toxic effects of saline NH_4_^+^-N wastewater, with minimal disruption observed in photosynthesis, protein synthesis, and the antioxidant system. Unlike previous studies that have focused on microalgae for NH_4_^+^-N wastewater treatment, this study specifically aimed to investigate the feasibility of utilizing *Oocystis lacustris* for treating NH_4_^+^-N wastewater under high salinity conditions. These findings have the potential to broaden the horizons of research in microalga-based wastewater treatment technology and offer fresh perspectives for industrial wastewater treatment.

## Figures and Tables

**Figure 1 toxics-12-00353-f001:**
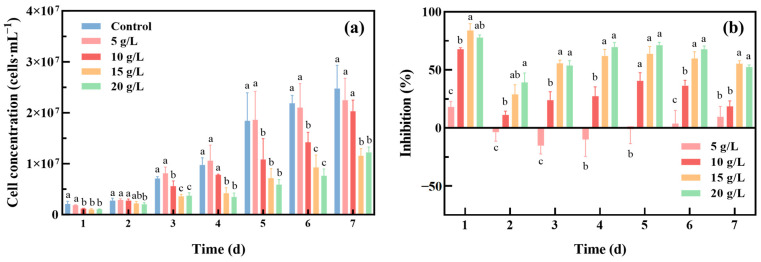
Growth of *Oocystis lacustris* at different Na_2_SO_4_ concentrations. (**a**) Cell concentration; (**b**) inhibition of yield. (At the same cultivation time, different letters on adjacent bars indicate significant differences (*p* < 0.05), while the same letter indicates no significant difference).

**Figure 2 toxics-12-00353-f002:**
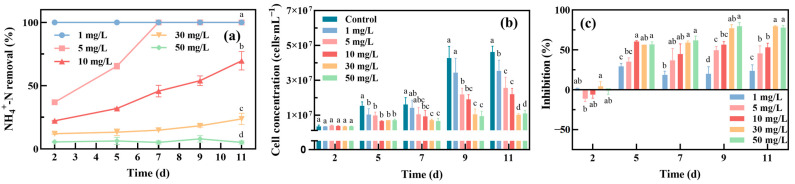
Treatment of *Oocystis lacustris* in the EX phase (8 × 10^5^ cells/mL) with different concentrations of NH_4_^+^-N. (**a**) NH_4_^+^-N removal rate; (**b**) cell concentration; (**c**) inhibition of yield. (At the same cultivation time, different letters on adjacent bars indicate significant differences (*p* < 0.05), while the same letter indicates no significant difference).

**Figure 3 toxics-12-00353-f003:**
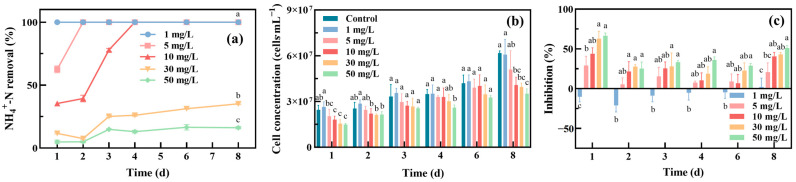
Treatment of *Oocystis lacustris* in the STA phase (2 × 10^7^ cells/mL) with different concentrations of NH_4_^+^-N. (**a**) NH_4_^+^-N removal rate; (**b**) cell concentration; (**c**) inhibition of yield. (At the same cultivation time, different letters on adjacent bars indicate significant differences (*p* < 0.05), while the same letter indicates no significant difference).

**Figure 4 toxics-12-00353-f004:**
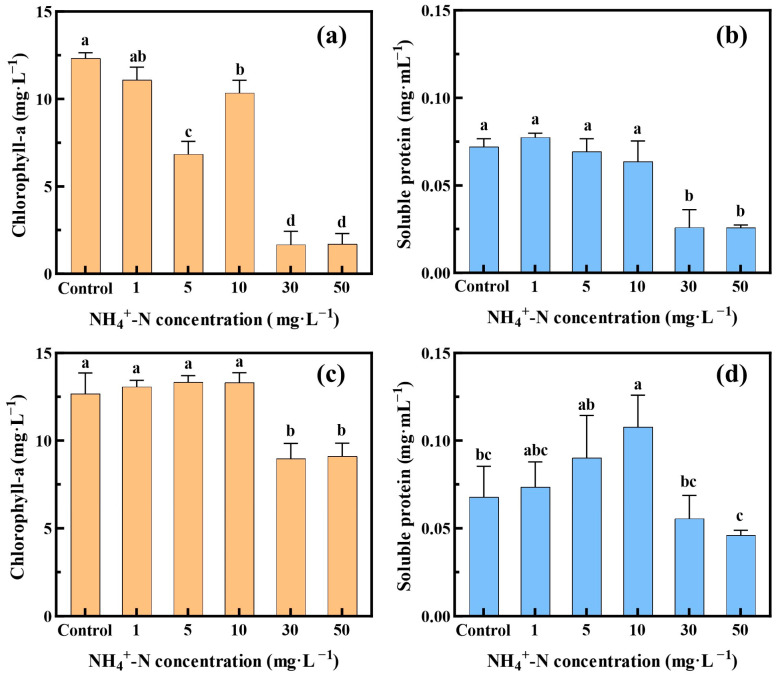
Variations in chlorophyll-a and soluble protein contents within *Oocystis lacustris* cells at different NH_4_^+^-N concentrations on the 8th day. (**a**,**b**) Changes in chlorophyll-a and soluble protein contents within algal cells in the EX phase; (**c**,**d**) changes in chlorophyll-a and soluble protein contents within algal cells in the STA phase. (Different letters on adjacent bars indicate significant differences (*p* < 0.05), while the same letter indicates no significant difference).

**Figure 5 toxics-12-00353-f005:**
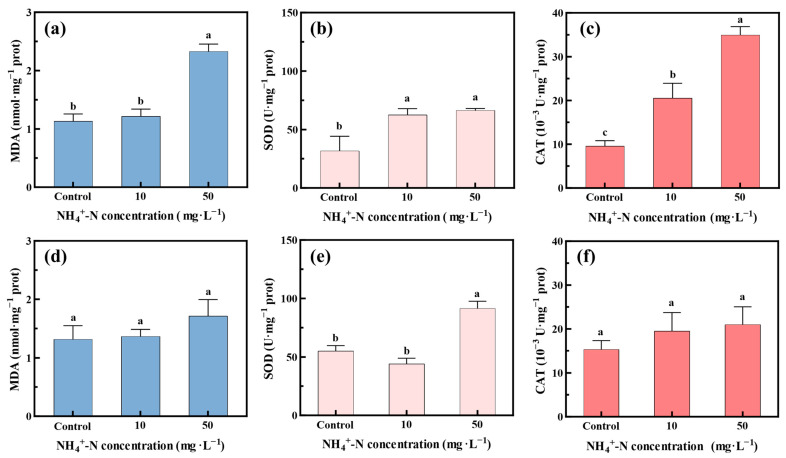
Oxidative stress status within *Oocystis lacustris* cells treated with different concentrations of NH_4_^+^-N on the 8th day. (**a**–**c**) MDA concentration and SOD and CAT activities within algal cells in the EX phase; (**d**–**f**) MDA concentration and SOD and CAT activities within algal cells in the STA phase. (Different letters on adjacent bars indicate significant differences (*p* < 0.05), while the same letter indicates no significant difference).

**Figure 6 toxics-12-00353-f006:**
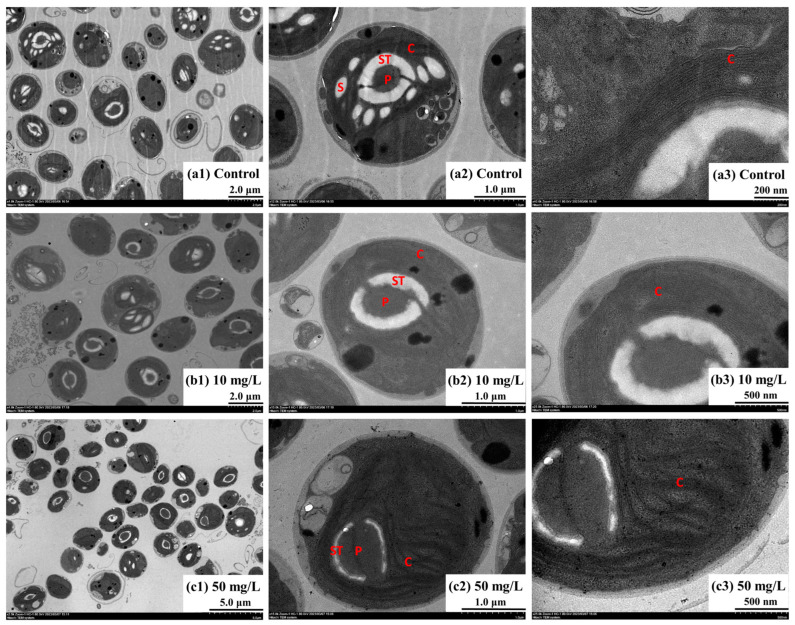
Microscopic morphology of *Oocystis lacustris* cells in the EX phase on the 8th day. (**a1**–**a3**) Control group; (**b1**–**b3**) 10 mg/L NH_4_^+^-N treatment group; (**c1**–**c3**) 50 mg/L NH_4_^+^-N treatment group. (C: chloroplast; ST: starch sheath; P: pyrenoid; S: starch grains).

**Figure 7 toxics-12-00353-f007:**
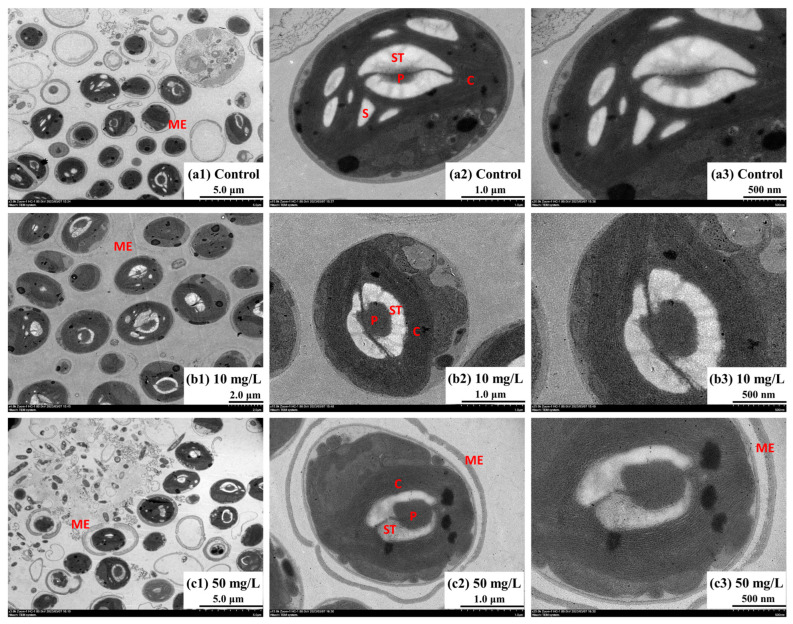
Microscopic morphology of *Oocystis lacustris* cells in the STA phase on the 8th day. (**a1**–**a3**) Control group; (**b1**–**b3**) 10 mg/L NH_4_^+^-N treatment group; (**c1**–**c3**) 50 mg/L NH_4_^+^-N treatment group. (C: chloroplast; ST: starch sheath; P: pyrenoid; S: starch grains; ME: mucilage envelope).

**Figure 8 toxics-12-00353-f008:**
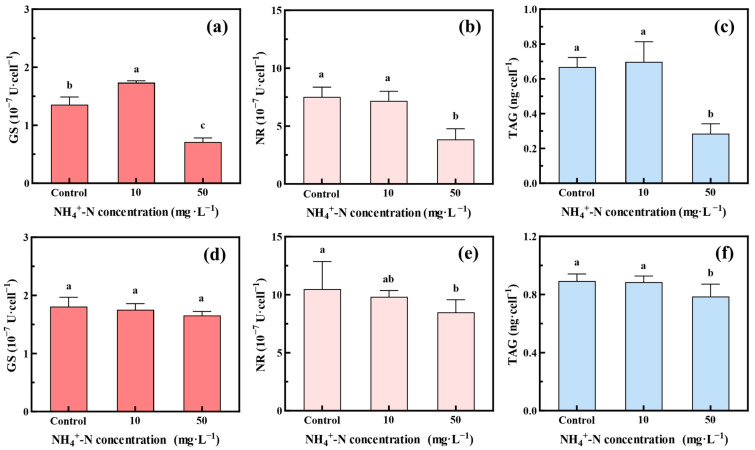
Changes in nitrogen metabolism enzymes and TAG activity within *Oocystis lacustris* treated by different NH_4_^+^-N concentrations on the 8th day: (**a**) GS, (**b**) NR, and (**c**) TAG within algal cells in the EX phase; (**d**) GS, (**e**) NR, and (**f**) TAG within algal cells in the STA phase. (Different letters on adjacent bars indicate significant differences (*p* < 0.05), while the same letter indicates no significant difference).

**Table 1 toxics-12-00353-t001:** The comparison of NH_4_^+^-N treatment efficiency by typical microalgae.

Algal Strain	Initial NH_4_^+^-N Concentration (mg/L)	NH_4_^+^-N Removal Efficiency (%)	Salt Concentration	Wastewater	Ref.
*Chlorella* sp.	85.9 ± 1.1	93.9	/	Municipal wastewater	[38]
*Chlorella vulgaris*	80	85.30	NaCl = 64 mg/L	Synthetic wastewater	[39]
*Chlorococcum* sp. GD		91.72			
*Parachlorella kessleri* TY		92.68			
*Scenedesmus obliquus*		93.04			
*Scenedesmus quadricauda*		97.03			
*Chlorella vulgaris*	10	53.12	/	Modified Bristol medium	[40]
*Scenedesmus* sp. LX1	15	31.1	/	Synthetic wastewater	[41]
*Scenedesmus quadricauda*	42~46	65	/	Synthetic sewage	[42]
*Chlorella* sp.	160	50.60	/	Mixed wastewater	[43]
*Oocystis lacustris*	1~50	15.2~99	Na_2_SO_4_ = 10 g/L	Modified BG11 medium	This work

## Data Availability

Data are contained within the article.

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
