# Peer review of "Stress Responses and Ammonia Nitrogen Removal Efficiency of *Oocystis lacustris* in Saline Ammonium-Contaminated Wastewater Treatment"

_toxics, 2024, doi:10.3390/toxics12050353_

Round 1

Reviewer 1 Report

Comments and Suggestions for Authors

The review topic is interesting and the manuscript is well written. Therefore, the manuscript has some problems that are listed below:

1) Please, add more results and mention the impact of your study in this research field.

2) The authors should contextualize the reason for using Oocystis lacustres. Which region is it native to? Please, add information on real wastewater (textile industry) containing both salinity and NH4 and their concentration.

3) What is the novelty of your study? Please add this information in the last paragraph of the introduction.

4) The authors should explain the choice of using the range of salinity (section 2.3) and ammonia (section 2.4) tested.

5) Section 2.2. Which pH was the microalgae cultivated? It affects the ammonia form in the solution.

6) The discussion is poor. The authors should compare the results found here with the literature. A table comparing all the effects with the literature is therefore recommended.

7) The conclusion should be more concise. What is the impact of your study in this research field?

Author Response

Dear reviewer,

Thank you for your evaluation and suggestions regarding our manuscript. Please find our point-by-point response in the attachment.

Best regards, 

Dr. Chensi Shen

College of Environmental Science and Engineering,

Donghua University, Shanghai 201620, China

Tel./Fax: +86-21-6779-2523, E-mail: shencs@dhu.edu.cn.

Reviewer 2 Report

Comments and Suggestions for Authors

The science quality is generally good.  The paper suffers from being China-centric.  Eutrophication was not first published in China.  You cite original work to demonstrate you understand the literature, versus self citing.  Eutrophication was recognized in the 19402. Look at Oliviera and Machado 2013 Env Techn Rev 2(1):117-127 for an introduction.  This paper lscks a quality review in introduction for citing non Chinese references to gage their work. Citations 1,2 are a joke compared to what has been published on the subject.

Second, Na versus Cl... Salinity is measured as Cl units, not Na.  I understand why you used Na, but it is not strictly correct to consider it Cl equivalents or salinity.  Ecologically what is the effect of Na on plants? Read..I EXP BOT

 2014 Mar;65(3):849-58.Sodium in plants: perception, signalling, and regulation of sodium fluxes

Page numbering is faulty

l48 sideline? better word??

l52 Currently, ...

l106: umol photons m2 s1 versus lux!

l107: agitation, how, rate of shaking?

l133: 1 to 1 Lugols would not be good! recheck

l137: this was not your invention, cite the original toxicological source, not your chinese text that stole the credit

l141: [26] wrong citation

l143: [27] wrong citation.. acetone or methanol extration, spec, details

l150 describe fixation/prep

l153: such as?? you serious!  not credible as written. Significance level, etc lacking here.

Fig 1: use asterisks to indicate significant differences

l178: end of last word missing?        lastly results and discussion combined. l205, 210, 190,228, etc  Comments on the Quality of English Language

significance implies statistical difference.. notable is not a sicence word of choice

Round 2

Reviewer 1 Report

Comments and Suggestions for Authors

The quality of the manuscript has been improved.

Author Response

Once again, thank you for your thorough review of our manuscript and for providing helpful suggestions.

Reviewer 2 Report

Comments and Suggestions for Authors

Your introductory phrasing on dye use of sulfates is highly repetitive 

Figs 1b,2,3 have no significant differences??

Comments on the Quality of English Language

ok

Author Response

Thank you for your time in reviewing this manuscript. Kindly find attached the point-by-point responses.
